# PLP-NER: Point-Line-Plane Fusion for Named Entity Recognition with BERT

## Abstract

Current state-of-the-art Named Entity Recognition systems commonly leverage an architecture that integrates BERT with Conditional Random Fields. Nevertheless, BERT is inherently constrained in capturing comprehensive global contextual semantics due to its Masked Language Modeling pre-training objective. To address this limitation, A novel "point–line–plane" contextual fusion framework is proposed. Within this paradigem, the [CLS] token functions as a "plane" that provides a compressed global representation, while the attention weights between the [CLS] token and individual tokens form a "line", which captures semantic topological relationships. These multi-grained features are subsequently incorporated into token representations via a Graph Neural Network, considerably enriching their contextual expressiveness. Furthermore, we introduce a Dynamic Linear-Chain CRF that adaptively models label transitions using attention-mechanized probability estimates, thereby overcoming the inflexibility of conventional CRFs. Extensive experiments on multiple benchmark datasets demonstrate that our approach consistently and significantly surpasses competitive baselines, achieving a notable 3.91 point gain in F1-score.

## 1 INTRODUCTION

Named Entity Recognition (NER) is a core task in natural language processing (NLP) that identifies and classifies named entities such as people, organizations, and locations in unstructured text (Nadeau & Sekine (2007)). As a fundamental component of the NLP pipeline, NER underpins a wide range of downstream applications, including information extraction, question answering, and knowledge graph construction. The task involves two key steps: entity span detection, which identifies the boundaries of an entity, and entity type classification, which assigns its semantic category (Luo et al. (2019)).

Early NER research relied on rule-based and statistical learning methods (Grishman & Sundheim (1996), Nadeau & Sekine (2007)). While effective in their time, these approaches were limited by their reliance on handcrafted features and shallow semantic understanding (Augenstein et al. (2017), Bengio et al. (2003)). The field was revolutionized by the advent of Transformer-based pre-trained language models (PLMs), such as BERT (Vaswani et al. (2017), Devlin et al. (2019)) These models leverage self-attention mechanisms to generate rich, contextualized representations, significantly advancing the state of the art in NER. Architectures that combine PLMs with a Conditional Random Field (CRF) layer, like BERT-CRF, have become standard baselines, marrying the semantic power of Transformers with the global sequence decoding capabilities of CRFs (Devlin et al. (2019),Huang et al. (2015)).

Despite their success, current PLM-based models for NER face several notable challenges. The pre-training objective, typically Masked Language Modeling (MLM), can lead to incomplete contextual semantics for the fine-grained NER task(Meng et al. (2024)). Furthermore, the pretraining–finetuning divergence introduces a representation shift that can degrade model performance and generalization (Villena et al. (2024), Cui et al. (2020)). Finally,

while CRFs enhance decoding, their static transition structure is rigid and fails to adapt to the specific semantic correlations present in different input sentences (Wang & Ji (2022)).

To address these limitations, we propose a novel Point–Line–Plane Fusion Framework for NER (PLP-NER). Our framework is designed to enrich semantic representation, improve structural adaptability, and enhance model generalization. Our key contributions are as follows:

- Multi-level Contextual Fusion: 1. We introduce a point–line–plane mechanism that effectively integrates token-level, pairwise, and global representations, enriched by graph neural networks, to capture more comprehensive semantic relationships.
- Dynamic Sequence Decoding: We design a dynamic linear-chain CRF that computes input-specific transition probabilities, enabling more flexible and context-sensitive label prediction.
- Masking-based Embedding Strategy:3.We employ a novel masking mechanism during training to mitigate the pretraining–finetuning divergence, thereby improving the model's robustness and generalization.

Extensive experiments on multiple benchmark datasets demonstrate that our framework consistently outperforms strong baselines, achieving an average F1-score improvement of 3.91 points. These results confirm the effectiveness of our approach in pushing the boundaries of NER performance.

Experimental results on multiple standard NER benchmarks demonstrate that the proposed contextual information fusion mechanism consistently achieves significant performance improvements over current SOTA methods. Specifically, our approach yields F1-score gains of 3.91 percentage points across several datasets, comprehensively validating its effectiveness in enhancing model performance and robustness, as well as advancing capabilities in semantic representation, structural adaptability, and generalization.

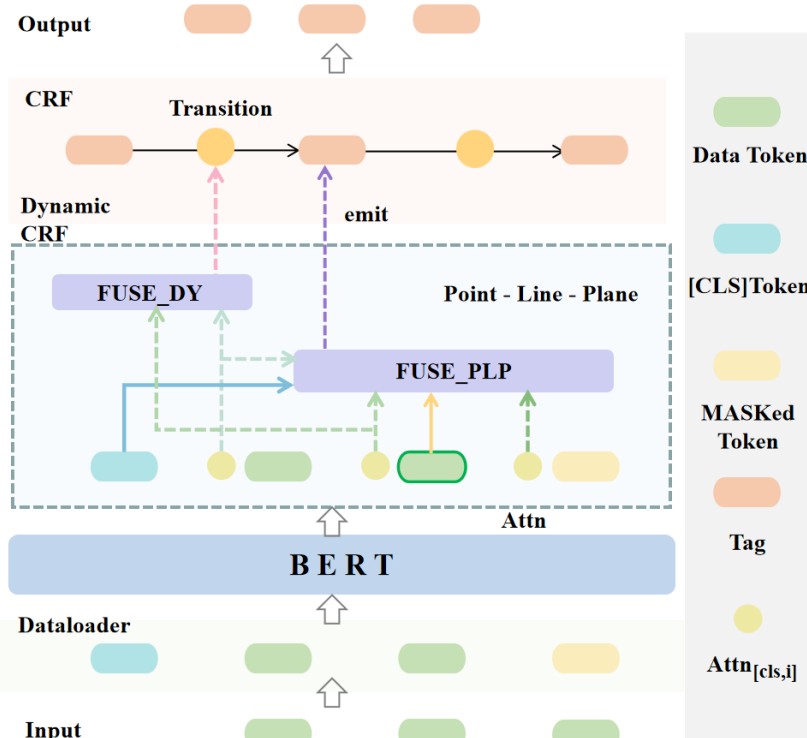

Figure 1: Point-Line-Plane Fusion Frame Diagram

## 2  RELATED WORK

The field of Named Entity Recognition (NER) has undergone a significant evolution, progressing from early rule-based systems to sophisticated deep learning and pre-trained models.

### 2.1  Main Approaches In NER.

#### 2.1.1  Rule-based and Statistical Models.

Early NER systems were largely based on handcrafted rules, dictionaries, and lexicons (Grishman & Sundheim (1996)). While precise for specific domains, these methods were labor-intensive to develop, brittle, and lacked the generalization capacity needed for diverse text. The field subsequently moved toward statistical models, which framed NER as a sequence labeling task. Probabilistic models like Hidden Markov Models (HMMs) (Rabiner (1989)), Maximum Entropy Models (MEMs) (Berger et al. (1996)), and, most notably, Conditional Random Fields (CRFs) (Lafferty et al. (2001)) became standard. CRFs were particularly effective due to their ability to model global dependencies and avoid the strong independence assumptions of HMMs. However, these models were still limited by their reliance on shallow, manually engineered features and struggled to capture long-range semantic context.

#### 2.1.2  Neural Representation Learning

The advent of deep learning introduced a paradigm shift by enabling models to learn features automatically. Recurrent Neural Networks (RNNs), particularly Long Short-Term Memory (LSTM)(Hochreiter & Schmidhuber (1997)) networks, proved adept at capturing sequential dependencies. The BiLSTM-CRF architecture (Huang et al. (2015)) became a popular and powerful model, combining the sequential feature learning of a bidirectional LSTM with the global sequence decoding of a CRF. This combination improved consistency in label predictions and handled out-of-vocabulary words more robustly. Other neural architectures, such as Convolutional Neural Networks (CNNs) (Ma & Hovy (2016)), were also used to extract local features from character and word embeddings. Despite these improvements, these models still faced limitations in capturing global, document-level context due to the nature of their sequential processing.

#### 2.1.3  Pre-trained Language Models (PLMs)

The most significant recent breakthrough in NER has been the adoption of large-scale pre-trained language models (PLMs) based on the Transformer architecture (Vaswani et al. (2017)). Models like BERT (Devlin et al. (2019)), RoBERTa (Liu et al. (2019)), and Span-BERT (Joshi et al. (2020)) are pre-trained on massive text corpora to learn deep, contextualized representations, which can then be fine-tuned for downstream tasks like NER. The standard BERT-CRF architecture, which uses BERT as an encoder to produce rich contextual embeddings and a CRF layer for structured prediction, has become the dominant discriminative approach in the field. These models have set new state-of-the-art results across a wide range of NER benchmarks.

### 2.2  Recent Trends and Limitations

While PLMs have propelled NER to new heights, several active research areas aim to address their remaining limitations. One direction is exploring generative approaches, which frame NER as a text-to-text task using models like T5 (Raffel et al. (2019)) or GPT (Brown et al. (2020)) (Yan et al. (2021)). These methods can handle complex nested and discontinuous entities but often come with high computational costs and remain less widely adopted than discriminative models. Another trend is retrieval-augmented NER, which uses external knowledge bases to enrich entity representations and handle low-resource or unseen entities (Lewis et al. (2020)).

Despite their effectiveness, current PLM-based models still face challenges related to semantic fusion and domain adaptability. Fine-tuning a pre-trained model on a new domain can lead to a pretraining–finetuning divergence, where the representations shift, hurting

performance (Liu et al. (2021)). Furthermore, the static nature of standard CRF transition probabilities limits their ability to capture fine-grained, input-specific semantic correlations. Our work builds upon the powerful BERT-CRF framework and introduces novel mechanisms to address these specific challenges through multi-level contextual fusion, dynamic decoding, and a masking-based training strategy.

## 3 METHODOLOGY

### 3.1 Preliminaries

NER is a sequence labeling task. Given an input sequence $X = (x_1, x_2, \ldots, x_n)$, the objective is to predict the corresponding label sequence $Y = (y_1, y_2, \ldots, y_n)$. Our approach is built upon the well-established BERT+CRF framework, which models the conditional probability of the label sequence $P(Y \mid X)$ by maximizing its likelihood. The training process involves minimizing the negative log-likelihood loss:

$$\mathcal{L}(\theta, \varphi, T) = -\log P(Y^* \mid X, \theta, \varphi, T) = -\left[\text{score}(Y^*, X; \theta, \varphi, T) - \log\left(\sum_{Y \in \mathcal{Y}(X)} \exp(\text{score}(Y, X; \theta, \varphi, T))\right)\right]$$

(1)

The sequence score is defined as the sum of emission and transition scores:

$$\text{score}(Y, X; \theta, \varphi, T) = \sum_{i=1}^{n} E_\varphi \left(B_\theta(x_i \mid X)\right) + \sum_{i=1}^{n-1} T_{y_i, y_{i+1}}$$

(2)

Here, $B_\theta(\cdot)$ denotes the token representation from BERT, $E_\varphi(\cdot)$ is the emission score function, and $T_{y_i, y_{i+1}}$ is the transition score from a static, learnable matrix $T$. This framework serves as our foundational baseline due to its effectiveness in balancing semantic representation and structured decoding.

### 3.2 Point–Line–Plane Contextual Fusion

While BERT+CRF is powerful, it suffers from insufficient contextual fusion. The representations from BERT, trained with objectives like MLM and NSP, often lack a comprehensive understanding of global semantic structure. To address this, we propose the Point–Line–Plane (PLP) Contextual Fusion mechanism, which draws an analogy from geometry to enhance structured semantic modeling.

- Semantic Points: We treat each token embedding, $B_\theta(x_i \mid X)$, as a semantic point, representing the local contextual semantics of an individual token.
- Semantic Plane: The [CLS] token, $B_\theta(\text{CLS} \mid X)$, serves as a semantic plane, providing a compressed representation of the global context.
- Semantic Lines: The attention weights between the [CLS] token and each word token, $a_\theta(x_i \mid X) = \text{Attn}_\theta(\text{CLS} \to x_i \mid X)$, are conceptualized as semantic lines. They explicitly capture the topological relationship between local and global contexts.

This framework can be viewed as a simplified Graph Neural Network (GNN) where the [CLS] node is a central hub for all other token nodes. Our approach enhances the emission score function $E_\varphi$ of the CRF to incorporate these multi-level representations, resulting in a new functional form:

$$E_i = f\left(B_\theta(x_i|X), B_\theta(\text{CLS}|X), a_\theta(x_i|X)\right)$$

(3)

This function enriches each token's representation with global and structural information before it is passed to the CRF. We implement this fusion using a two-stage Multi-Layer Perceptron (MLP):

$$f(\cdot) = \text{MLP}\left(\text{MLP}\left(B_\theta(\text{CLS}|X) \oplus a_\theta(x_i|X)\right) \oplus B_\theta(x_i|X)\right)$$

(4)

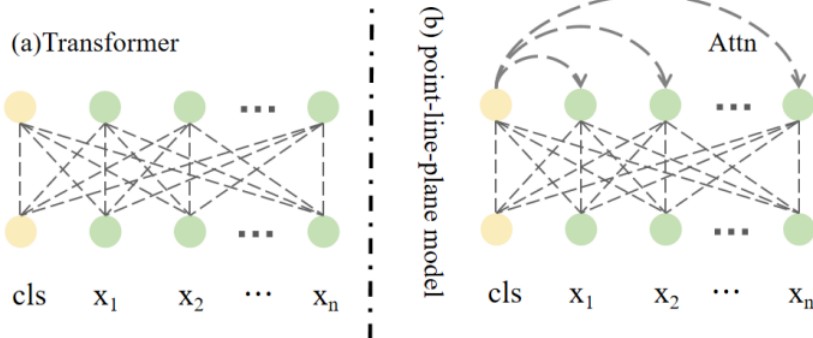

Figure 2: Geometric schematic diagram of Point-Line-Plane integration modeling based on transformer architecture.

where $\oplus$ denotes vector concatenation. This two-stage process ensures a deeper integration of global and structural information into the token-level representations.

To further enhance the robustness of boundary detection, we introduce Neighborhood Enhancement based on our observation that attention weights between adjacent tokens and the [CLS] token provide strong cues for entity boundaries and label consistency. We modify the fusion function to incorporate these neighborhood features:

$$f(\cdot) = \text{MLP}\left(\text{MLP}\left(B_\theta(\text{CLS}|X) \oplus a_\theta(x_{i-1} : x_{i+1}|X)\right) \oplus B_\theta(x_i|X)\right) \tag{5}$$

This new function explicitly leverages local interaction patterns to inform the model, enhancing its ability to handle complex entity boundaries.

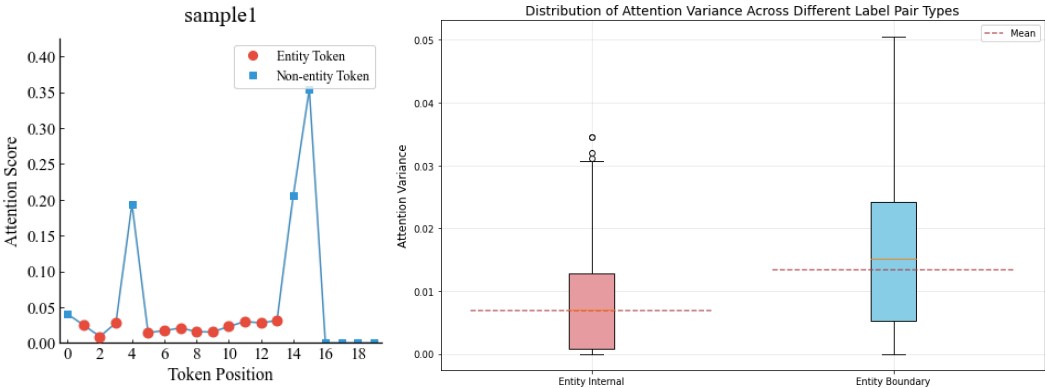

Figure 3: Statistical Results of Differences in Attention Scores Between Entity Boundary and Internal Tokens Towards the CLS Token and Sample Line Chart Display.

### 3.3 Dynamic Linear-Chain CRF

Standard CRFs use a static transition matrix $T$ that encodes global, corpus-level statistics. This rigidity makes it difficult to adapt to a specific sentence's context, leading to suboptimal predictions for complex or ambiguous transitions. To overcome this, we propose a Dynamic Linear-Chain CRF that modifies transition scores based on local, input-specific features.

Our attention visualization analysis reveals that attention patterns show high consistency within entities and sharp changes at boundaries (see Figure 3). Motivated by this, we use the attention scores of adjacent tokens to the [CLS] token as features for dynamic transition adjustments. We define a mapping function $g_\beta : \mathbb{R}^d \to \mathbb{R}^3$ to produce a 3-dimensional vector $v_i = \left(g_\beta^{\text{in}}(s_i), g_\beta^{\text{bd}}(s_i), g_\beta^{\text{out}}(s_i)\right)$, where $s_i = [\text{Attn}(x_i, \text{cls}|X), \text{Attn}(x_{i+1}, \text{cls}|X)]$. These

components are designed to correct transitions within entities ($g^{\mathrm{in}}$), across entity boundaries ($g^{\mathrm{bd}}$), and between non-entities ($g^{\mathrm{out}}$).

To apply these corrections, we define a label transition index function $\kappa : Y \times Y \to \{0, 1, 2\}$, which assigns an index to each type of transition (e.g., 'B-PER' to 'I-PER' is an 'in-entity' transition). The dynamic transition score is then defined as:

$$D_\beta(y_i, y_{i+1}) = T_{y_i y_{i+1}} + \sum_{k=0}^{2} v_{i,k} \cdot \mathbb{I}\left[\kappa(y_i, y_{i+1}) = k\right] \tag{6}$$

This function modifies the base transition score $T_{y_i y_{i+1}}$ with a context-dependent correction $v_{i,k}$. We apply symmetric clipping to keep these dynamic adjustments within a small, stable range, preventing the model from becoming unstable while still allowing for fine-grained adjustments.

The overall sequence score becomes:

$$\mathrm{score}(Y, X) = \sum_{i=1}^{n} e_{i,y_i} + \sum_{i=1}^{n-1} D_\beta(y_i, y_{i+1}) \tag{7}$$

This hierarchical approach of "feature encoding - index mapping - dynamic correction" ensures that our model can adapt its transition probabilities to the specific context of each sequence, significantly improving its generalization capacity.

### 3.4 Training Objective

To mitigate the pretraining–finetuning mismatch and improve the robustness of our model, we incorporate a second objective. Similar to BERT's original training, we add an auxiliary Masked Language Modeling (MLM) loss. During training, we randomly mask 15% of the input tokens and train the model to reconstruct them. This auxiliary task forces the model to maintain its pre-trained semantic understanding, reducing the representation shift that often occurs during fine-tuning. Our final optimization objective is a combination of the primary NER loss and the auxiliary MLM loss:

$$\mathcal{L}(\theta, \varphi, \beta, \mathrm{T}) = \mathcal{L}_{\mathrm{ner}}(\theta, \varphi, \beta, \mathrm{T}) + \mathcal{L}_{\mathrm{mlm}}(\theta) \tag{8}$$

This joint training strategy leverages the best of both worlds, ensuring that the model remains sensitive to fine-grained token-level semantics while optimizing for the primary NER task.

## 4 EXPERIMENT

### 4.1 Datasets

To rigorously evaluate the effectiveness and generalization of the proposed method, we conduct experiments on four representative Chinese and English NER benchmarks—specifically including CoNLL2003 (English general-domain), WNUT17 (English low-resource), MSRA (Chinese general-domain), and CLUENER (Chinese domain-specific)—which span general-domain, low-resource, and domain-specific settings.

### 4.2 Implementation Details

We adopt the F1 score as the evaluation metric to assess model performance, defined as follows:

$$\mathrm{F1} = 2 \times \frac{\mathrm{Precision} \times \mathrm{Recall}}{\mathrm{Precision} + \mathrm{Recall}}, \mathrm{Precision} = \frac{TP}{TP + FP}, \mathrm{Recall} = \frac{TP}{TP + FN}$$

where $TP$, $FP$, and $FN$ denote true positives, false positives, and false negatives, respectively.

All experiments are implemented based on the BERT+CRF framework. We utilize the Hugging Face Transformers library to load pretrained models and tokenizers. Training is conducted on two NVIDIA GPUs, with the core hyperparameters and configurations summarized as follows:

Table 1: Statistics of Benchmark Datasets for NER Evaluation

| Dataset | Language | Train/Dev/Test | Entity Categories |
|---------|----------|----------------|-------------------|
| CoNLL-2003 | English | 14,987 / 3,466 / 3,684 | Person,Organization,Location, MISC |
| WNUT2017 | English | 1,000 / 128 / 1,283 | Person,Location,Organization, Product,Event,Corporation |
| MSRA | Chinese | 46,364 / - / 4,365 | Person,Location,Organization |
| CLUENER | Chinese | 10,748 / 1,343 / 1,345 | Address,Book,Company,Game, Government,Movie,Name, Organization,Position,Scene |

Table 2: Hyperparameter Settings

| Parameter | Value |
|-----------|-------|
| Maximum sequence length | 512 |
| Batch size per GPU | 12 |
| Optimizer | AdamW |
| Learning rate for BERT backbone | 3e-5 |
| Learning rate for dynamic CRF layer | 1e-3 |
| Learning rate for masking task | 1e-3 |
| Maximum training epochs | 10 |

### 4.3 Results And Analyse

Table 3: Experimental Results on CoNLL-2003, WNUT-2017, MSRA, and CLUENER

| Dataset/F1 | CoNLL2003 | WNUT2017 | MSRA | CLUENER |
|------------|-----------|----------|------|---------|
| SOTA | | | | |
| BERT+MRC+DSC(Li et al. (2020)) | 93.95 | - | 96.72 | 77.56 |
| ACE+document-context(Wang et al. (2020)) | 94.60 | - | - | - |
| W2NER(Li et al. (2021)) | 93.07 | - | 96.10 | - |
| Baseline | | | | |
| BERT+CRF | 93.95 | 60.14 | 94.41 | 80.76 |
| Ours | | | | |
| PLP-NER | 94.51 | 60.53 | 95.85 | 81.57 |
| +MASK | 94.60 | 61.09 | 96.15 | 82.27 |
| +NE | 94.93 | 61.39 | 96.93 | 84.12 |
| +DY | 95.07 | 60.77 | 96.89 | 84.67 |

We conducted a systematic comparison of our innovative model with baseline models, representative historical methods, and the current SOTA NER models. As shown in Table 3, we employ the macro-average F1 score on the test set as the quantitative evaluation metric for model performance. The experimental results demonstrate that the PLPA model consistently exhibits superior performance across all benchmark datasets.

It is noteworthy that the introduction of the Dynamic Linear-Chain CRF resulted in slight fluctuations in performance on the WNUT-17 dataset. This could be attributed to the relatively small size of the dataset, where the complex dynamic transition matrix increases the risk of overfitting. Nevertheless, our model still significantly outperforms the baseline methods under this setup. Particularly, on the fine-grained entity recognition benchmark CLUENER, the PLPA model achieves a substantial 3.91 percentage points improvement. This breakthrough can be attributed to two key factors: firstly, the current SOTA methods for this dataset still leave considerable room for improvement in fine-grained entity recognition; secondly, the attention-based scoring mechanism we propose effectively models the

boundary features of multi-class fine-grained entities, a task that existing methods struggle with due to their limited ability to perceive such complex boundary patterns.

Overall, a comprehensive analysis of the experimental results reveals that our proposed model demonstrates significant advantages in terms of generalization, robustness, and recognition accuracy. In particular, in the context of fine-grained entity recognition, the model's robust capability to model complex boundary patterns provides new insights and powerful tools for advancing NER technology, showcasing its considerable research value and application potential.

### 4.4 Ablation

As shown in Table 3, the stepwise ablation experiments verify the incremental contribution of each component:

- PLP-NER (base). Delivers consistent gains over BERT+CRF across four benchmarks (CoNLL-2003, WNUT-2017, MSRA, CLUENER), lifting F1 by +0.56/+0.39/+1.44/+0.81, indicating that point–line–plane fusion improves token-, span-, and structure-level interactions.
- +MASK. Provides further steady improvements of +0.09/+0.56/+0.30/+0.70 over PLP-NER, mitigating the train–test mismatch and enhancing generalization.
- +NE. Yields the largest incremental gains, especially on MSRA (+0.78) and CLUENER (+1.85), highlighting the importance of local context for boundary detection.
- +DY. Achieves the best overall F1 on CoNLL-2003 (95.07), MSRA (96.89), and CLUENER (84.67); on WNUT-2017, +DY is slightly lower than +NE (60.77 vs. 61.39) yet remains +0.63 above BERT+CRF.

Overall, the ablation trend suggests that (i) multi-granularity fusion (PLP-NER) establishes a strong foundation, (ii) regularized training (+MASK) yields stable gains, (iii) local context (+NE) is crucial for hard boundaries, and (iv) label-dependent dynamics (+DY) provide the final push to state-of-the-art performance on three datasets.

## 5 CONCLUSION

In this paper, we present an enhanced BERT-CRF framework that integrates semantic fusion, dynamic structural modeling, and training strategies, achieving significant gains in accuracy and robustness across multiple NER benchmarks.

### Acknowledgments

We used a large language model–based assistant during the final editing stage for language polishing (grammar, clarity, and style). The tool did not contribute to the research design, experiments, data analysis, or claims; the authors remain solely responsible for all content and any errors. We complied with ICLR's policy on responsible use of generative tools and manually verified outputs to avoid inclusion of unauthorized or sensitive material.

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
