# OpenReview forum: "PLP-NER: Point-Line-Plane Context Fusion for Named Entity Recognition"
_ICLR.cc/2026/Conference — ICLR 2026 Conference Withdrawn Submission_

### Official Review · Reviewer_ib8q · 2025-10-31

**Soundness:** 3
**Presentation:** 2
**Contribution:** 3
**Rating:** 4
**Confidence:** 4

**Summary:**

The paper introduces PLP-NER, an extension of the standard BERT-CRF framework for named entity recognition.
The main idea is to enhance contextual representations through a point–line–plane fusion mechanism, where token embeddings (“points”), [CLS]–token attention weights (“lines”), and the [CLS] representation (“plane”) are integrated via a multi-layer perceptron.
A second contribution is a Dynamic Linear-Chain CRF, which adjusts label transition probabilities based on input-specific attention features.
In addition, a masking-based auxiliary task is incorporated to improve model robustness and generalization.
Experiments on four datasets (CoNLL-2003, WNUT-2017, MSRA, and CLUENER) show moderate F1-score improvements over a BERT+CRF baseline. However, the empirical analysis, comparison breadth, and interpretability of the results remain limited.

**Strengths:**

* The paper addresses known limitations of BERT+CRF models in global semantic fusion and static transition modeling through a well-defined architectural extension.
* The proposed dynamic transition mechanism is conceptually reasonable and could potentially capture input-dependent label correlations.
* The modular design (PLP fusion + dynamic CRF) is clear and may be transferable to other sequence labeling tasks.

**Weaknesses:**

### 1. Writing and Presentation Issues

* The paper appears hastily prepared and insufficiently polished.

  * The final two paragraphs of the Introduction are redundant and lack clear logical progression.
  * Several abbreviations are undefined (e.g., NSP, PLPA, NE, DY).
  * Figures and tables are included but not well integrated or discussed, reducing clarity.
  * The overall structure and formatting show inconsistencies, suggesting limited proofreading.

### 2. Technical and Methodological Limitations

* The proposed Point–Line–Plane (PLP) fusion mainly combines existing BERT representations ([CLS], token embeddings, and attention weights) through simple concatenation and MLPs.
  While the geometric analogy is intuitive, the method is technically incremental, resembling prior work on attention pooling, [CLS]-based aggregation, and graph-enhanced contextual modeling for NER.
* The Dynamic Linear-Chain CRF is not clearly differentiated from previous Neural or Adaptive CRF variants that also incorporate input-dependent transitions.
  The paper lacks analysis of model stability, computational cost, or parameter sensitivity.
* Key implementation details—such as MLP layer count, hidden dimensionality, activation functions, and clipping thresholds—are missing, making the work difficult to reproduce and evaluate fairly.

### 3. Empirical Weaknesses

* The fine-grained NER results (e.g., on CLUENER) are under-analyzed. The paper claims significant improvements but provides no qualitative examples, case studies, or visualizations to explain the model’s advantages on fine-grained entity boundaries.
* The ablation design is questionable: experiments follow an additive scheme (+MASK → +NE → +DY) rather than removing components individually, making it unclear which component contributes most.
* The baseline comparisons rely on relatively outdated models (mostly 2019–2021), and the reference performance for WNUT-2017 and CLUENER is sparse, limiting the interpretability of reported gains.
* There is no statistical analysis (e.g., variance, confidence intervals, or significance tests) to verify that the observed F1 improvements are robust.

### 4. Insufficient Literature Review

The Related Work section lacks sufficient coverage of two major research directions:

1. Contextual fusion and representation aggregation using [CLS] embeddings, attention pooling, or graph-based architectures for NER;
2. Neural or adaptive CRF approaches that already incorporate dynamic or learnable transition potentials.
   These omissions make it difficult to position PLP-NER relative to existing research and to assess its genuine novelty.

**Questions:**

See the Weaknesses section for detailed methodological and empirical concerns.

---

### Official Review · Reviewer_sP9z · 2025-10-31

**Soundness:** 2
**Presentation:** 2
**Contribution:** 2
**Rating:** 2
**Confidence:** 5

**Summary:**

This paper proposes PLP-NER, a BERT-CRF–based NER model that fuses multi-granularity context via a “Point–Line–Plane” scheme: token embeddings (point), [CLS]→token attention (line), and a global [CLS] vector (plane). These signals are injected into the emission scores through lightweight fusion; a Dynamic Linear-Chain CRF further adjusts transition scores at instance level using attention-derived cues (with coarse in/boundary/out categories). An auxiliary MLM loss is used during fine-tuning to mitigate representation drift. Experiments on CoNLL-2003, WNUT-2017, MSRA, and CLUENER report improvements over a BERT-CRF baseline; the paper highlights a “+3.91 F1” gain. The work aims to enhance global/structure awareness within a standard supervised NER pipeline.

**Strengths:**

1. This paper addresses limited global/structural awareness of PLMs for NER and the rigidity of a static CRF transition matrix.

2. the PLP fusion and dynamic CRF are easy to add to a common BERT-CRF stack and appear to give consistent gains on several datasets.

**Weaknesses:**

1. In current era, I’d expect zero/few-shot LLM NER or LLM-augmented baselines. Without that, the value of a small BERT-CRF variant isn’t convincing.

2. The “avg +3.91 F1” looks like a single-dataset bump; please show the exact averaging math or revise the claim.

3. Missing significance/key stats. No multi-seed runs, std/CI, or significance tests—hard to judge robustness.
4. Important training details (seeds, hyperparams, compute/time/memory), code/checkpoints are missing; hard to reproduce.

**Questions:**

See weakness

---

### Official Review · Reviewer_7g7f · 2025-11-01

**Soundness:** 1
**Presentation:** 1
**Contribution:** 1
**Rating:** 0
**Confidence:** 5

**Summary:**

The paper proposes PLP-NER. A “point–line– plane” contextual fusion framework is proposed.  It introduces a Dynamic Linear-Chain CRF that adaptively models label transitions using attention-mechanized probability estimates.

**Strengths:**

The paper’s claimed innovations lack substantive novelty; all components are combinations of existing techniques, and their necessity has not been rigorously validated.

**Weaknesses:**

1.	Lack of statistical significance testing: The reported F1 improvements are modest in some cases. Without significance tests, it’s unclear whether gains are reliable, especially on smaller datasets like WNUT17.
2.	Limited comparison to recent SOTA: The “SOTA” row in Table 3 includes methods from 2020–2021, but omits more recent works. The claim of “surpassing competitive baselines” would be stronger with up-to-date comparisons.
3.	Overstated novelty of “GNN” interpretation: The paper describes the PLP fusion as a “simplified GNN,” but no actual message-passing or graph propagation occurs.
4.	Inconsistent and Confusing Nomenclature: The method is introduced as PLP-NER throughout the abstract, introduction, and methodology, but suddenly referred to as “PLPA” in Section 4.3
5.	Poorly Constructed Tables: Table 3 is disorganized and lacks clear column alignment.
6.	The text in paper contains redundant and circular claims.
7.	Uninformative and Misleading Figures:Figure 1 is a generic block diagram with no architectural detail.Figure 2 claims to be a “geometric schematic” but merely overlays geometric terms on a Transformer diagram without formalizing the analogy.

**Questions:**

8.	GNN claim clarification: The paper states the PLP framework “can be viewed as a simplified Graph Neural Network.” Could the authors clarify what graph structure is assumed and how message passing is implemented? If no inter-token messages are exchanged, is the GNN analogy purely conceptual?
9.	Statistical significance: Were the reported F1 scores averaged over multiple random seeds?
10.	Attention source specification: Which attention weights are used in the “semantic lines”? Are they from the last BERT layer? Averaged over heads?
11.	Generalization beyond BERT: Has the PLP framework been tested with other PLMs (e.g., RoBERTa, DeBERTa)? If the gains are consistent, it would strengthen the claim that the method addresses a general limitation of PLM-based NER, not just BERT-specific artifacts.
12.	Dynamic CRF overfitting analysis: For WNUT17, the Dynamic CRF (+DY) lags behind +NE. Did the authors observe overfitting during training

---

### Official Review · Reviewer_d7qa · 2025-11-01

**Soundness:** 2
**Presentation:** 3
**Contribution:** 1
**Rating:** 2
**Confidence:** 4

**Summary:**

The paper **“PLP-NER: Point-Line-Plane Fusion for Named Entity Recognition with BERT”** introduces a novel framework enhancing BERT-CRF models through geometric contextual fusion and adaptive decoding. By treating tokens as “points,” attention relations as “lines,” and the [CLS] token as a “plane,” the model fuses multi-level semantics via a graph-based mechanism, while a **Dynamic Linear-Chain CRF** adapts label transitions contextually. A masking-based auxiliary loss further mitigates pretraining–finetuning divergence. Experiments on four NER benchmarks show consistent gains—up to **+3.9 F1 points**—especially in fine-grained recognition. Overall, it’s a well-motivated and methodologically solid contribution, though computational efficiency analysis is limited.

**Strengths:**

* **Clear Motivation:** The paper clearly identifies key limitations in existing BERT-CRF NER systems—namely weak global contextualization and rigid decoding—and justifies the need for multi-level semantic fusion and dynamic transitions.
* **Novel Structure:** The proposed **Point–Line–Plane fusion** is an elegant and interpretable geometric abstraction that effectively integrates local, relational, and global information, while the **Dynamic Linear-Chain CRF** introduces adaptive, context-aware label transitions.
* **Sound Experimental Results:** Comprehensive evaluations across four diverse benchmarks (English and Chinese, general and domain-specific) demonstrate consistent and meaningful performance gains, including a notable +3.9 F1 improvement, supported by detailed ablations verifying each component’s contribution.

**Weaknesses:**

* **Limited Comparison Scope:** While the paper thoroughly benchmarks against traditional BERT-CRF and recent discriminative NER models, it omits comparisons with **decoder-only large language models (LLMs)** such as GPT/Qwen generative NER frameworks etc. Given the growing relevance of generative and instruction-tuned models for sequence labeling, such comparisons would strengthen the claim of state-of-the-art performance and clarify the advantages of the proposed fusion mechanism under modern NER paradigms.

**Questions:**

1. why not compare the proposed model's performance with decoder-only language models?

---

### Note · Authors · 2025-11-16

I have read and agree with the venue's withdrawal policy on behalf of myself and my co-authors.